# The deepwater oxygen deficit in stratified shallow seas is mediated by diapycnal mixing

Tom Rippeth [1] ✉, Sijing Shen [1], Ben Lincoln[1], Brian Scannell[1], Xin Meng[2], Joanne Hopkins [3] & Jonathan Sharples [2]

Seasonally stratified shelf seas are amongst the most biologically productive on the planet. A consequence is that the deeper waters can become oxygen deficient in late summer. Predictions suggest global warming will accelerate this deficiency. Here we integrate turbulence timeseries with vertical profiles of water column properties from a seasonal stratified shelf sea to estimate oxygen and biogeochemical fluxes. The profiles reveal a significant subsurface chlorophyll maximum and associated mid-water oxygen maximum. We show that the oxygen maximum supports both upward and downwards $O_2$ fluxes. The upward flux is into the surface mixed layer, whilst the downward flux into the deep water will partially off-set the seasonal $O_2$ deficit. The results indicate the fluxes are sensitive to both the water column structure and mixing rates implying the development of the seasonal $O_2$ deficit is mediated by diapcynal mixing. Analysis of current shear indicate that the downward flux is supported by tidal mixing, whilst the upwards flux is dominated by wind driven near-inertial shear. Summer storminess therefore plays an important role in the development of the seasonal deep water $O_2$ deficit.

The seasonally stratified shelf seas are disproportionately important in supporting oceanic primary production globally[1–3]. In consequence, they host important fisheries[4] and are a dynamic component of the global carbon cycle[5–7]. They are also viewed as increasingly important locations for the capture of wind energy, with a planned rapid expansion over the next decades driving large-scale developments of wind turbine infrastructure into the deep seasonally stratified areas[8].

Oxygen is fundamental to biological and biogeochemical processes in marine systems[9] with deficits in $O_2$ detrimental to marine life and biogeochemical cycling[10]. In seasonally stratified shelf seas a deep water oxygen deficit develops during the period of stratification, as bacteria break down upper water column-derived organic material that has sunk into the deep water. Climate simulations predict that this deficit will grow in response to a warming climate[11,12] as a result of changes in spring temperature and stratification strength and duration[13].

Seasonal stratification occurs in response to the annual variation in heat exchange at the sea surface. Whilst the water column remains well mixed during winter, stratification develops in the spring and persists through into the autumn. At this time, the seasonal thermocline acts as a physical barrier separating the surface mixed layer from the deep water, with exchange of heat and dissolved matter between these layers mediated by diapycnal mixing[14–18].

Biological productivity is tightly controlled by the timing and the strength of seasonal stratification[19,20]. The onset of stratification triggers the annual 'spring bloom' which persists until the supply of limiting nutrients (nitrate) in the surface mixed layer (SML) is exhausted. Following the spring bloom, primary production is sustained within the subsurface chlorophyll maximum (SCM) through the mixing up of limiting nutrients into the thermocline[14,15,17,21]. The SCM is estimated to account for approximately 50% of annual primary production in these regimes[22–24].

A mid-water $O_2$ maximum develops in response to the SCM primary production[25] with diapycnal mixing supporting substantial $O_2$ fluxes which ventilate the deep water[17,25]. Accordingly, over the summer, diapycnal mixing supplies limiting nutrients to support the

[1]School of Ocean Sciences, Bangor University, Anglesey LL59 5AB Wales, UK. [2]School of Environmental Sciences, University of Liverpool, Liverpool L69 3GP, UK. [3]Marine Physics and Ocean Climate, National Oceanography Centre, Liverpool L3 5DA, UK. ✉e-mail: t.p.rippeth@bangor.ac.uk

primary production within SCM, and simultaneously replenishing the $O_2$ deficit created by the remineralisation of sunken organic matter associated with the primary production[12,26,27].

Here we combine time series of turbulence measurements from the seasonally stratified Celtic Sea (Fig. 1), with water column profiles of physical and biogeochemical parameters to investigate the impact of diapycnal mixing on the deep-water oxygen budget over summer. We also identify the physical processes supporting the diapycnal mixing. We show that the diapycnal mixing results in fluxing of dissolved oxygen associated with the SCM into both the SML and the deep water, indicating a leakage of oxygen to the atmosphere. This has the potential to enhance the seasonal deep water oxygen deficit.

## Results

Data was collected over the summer of 2014 at a mooring deployed at a seasonally stratified location in the central Celtic Sea far from freshwater influence. They comprise turbulence ($\epsilon$ - the rate of dissipation of turbulent kinetic energy) time series made at 3 depths together with multilevel temperature and salinity time series. Full water column velocity profiles are obtained from a seabed-mounted acoustic Doppler current profiler (ADCP).

Stratification begins to form in early April as the surface layer warms in response to a positive heat input (Fig. 2). In late July the temperature difference between the surface mixed layer (SML) and deep water reaches a maximum of around 10 °C. Over this period the density structure is dominated by temperature with a negligible salinity difference between the surface and deep layers. The SML warms in response to strong surface heating, reaching a maximum temperature of 20 °C in early August at which time the SML is approximately 40 m deep. From early August the surface heat flux starts to reverse, the SML cool and the thermocline deepens. The stratification is finally destroyed towards the end of the year. Over the stratified period the deep water oxygen concentration ($O_2$) is observed to decline from around 280 m mol m$^{-3}$, at the onset of stratification, to around 235 m mol m$^{-3}$ in late November.

### Evolution of water column structure and mixing

Time series of the rate of dissipation of turbulent kinetic energy ($\epsilon$) at three fixed levels in the thermocline region are shown in Fig. 2d for June to August 2014. They show that the observed daily mean values of ($\epsilon$) vary in both time and with depth. For the initial part of the deployment all three instruments were situated within the thermocline. However, by mid-July the instrument at 16 m depth was in the SML. In June, the daily mean value of $\epsilon$ close to the base of the thermocline (47 m) was observed to be $0.3 \times 10^{-7}$Wkg$^{-1}$. Higher up in the thermocline interior (35 m), the observed values are a little higher. During July, $\epsilon$ in the thermocline interior (35 m) was $10^{-7}$Wkg$^{-1}$, whilst $\epsilon$ close to the thermocline base (47 m) varied between $0.5 - 1.0 \times 10^{-7}$Wkg$^{-1}$. In August $\epsilon$ in the thermocline interior (35 m) had increased further with two peaks of $3.2 \times 10^{-7}$Wkg$^{-1}$ and $2.5 \times 10^{-7}$Wkg$^{-1}$, respectively. The variability in $\epsilon$ at 47 m is less marked with a maximum of $2 \times 10^{-7}$Wkg$^{-1}$ in early August and subsequent decline (Fig. 2d). $\epsilon$ at 16 m depth are up to an order of magnitude larger than those observed in the thermocline interior during the first couple of weeks of the deployments, after which time that instrument had moved into the SML with larger values of $\epsilon$ observed.

Over the summer the deep water is observed to warm slowly, with the temperature rising by 0. 9 °C between June and September, Fig. 2b. The vertical temperature gradient over the thermocline and the $\epsilon$ values at 35 m depth are combined using the dissipation method[28] to give an average downward diapycnal heat flux of 40 W m$^{-2}$ over this period. This heating rate is sufficient to explain the observed deep water warming indicating that diapycnal mixing is the first order control on the heat flux to the deep water at this time. As vertical gradients in dissolved matter were very much stronger than horizontal

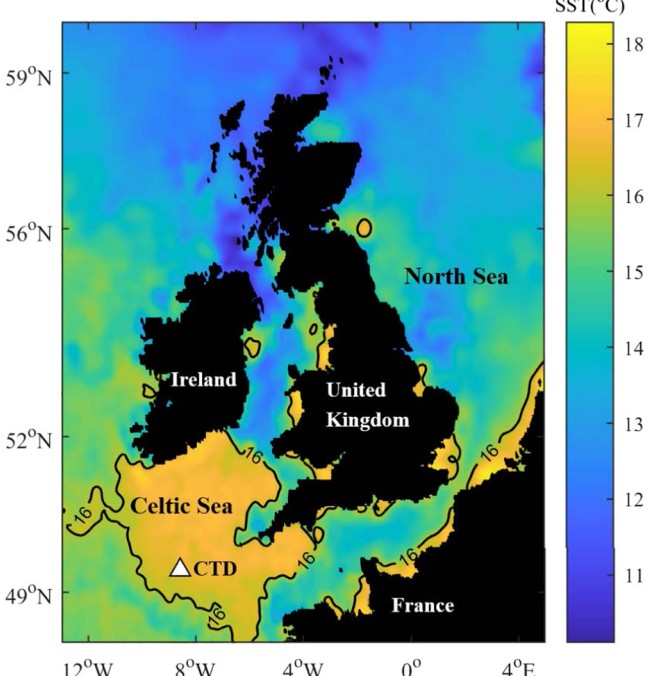

**Fig. 1 | A map showing the northwest European shelf seas on which the location of the measurements is shown as a △.** The map is contoured for daily averaged sea surface temperature at the beginning of the period of interest (19th June 2014). The areas with temperatures > 16°C are the seasonally stratified Celtic Sea. The sea surface temperature (SST) is downloaded from NERC Earth Observation Data Analysis and Artificial-Intelligence Service (NEODAAS) Plymouth Marine Laboratory (https://data.neodaas.ac.uk/visualisation/). Source data are provided as a Source Data file.

gradients and, during the period of interest, the residual flow is weak ($< 2c$ ms$^{-1}$)[29], it is reasonable to assume that diapycnal mixing will also dominate the transport of dissolved matter.

### Biogeochemical fluxes

Profiles of the vertical structure of the water column at the mooring location, including chlorophyll fluorescence (a proxy for phytoplankton biomass) and dissolved oxygen, together with discrete bottle samples of limiting nutrient NOx (NO$_3$ + NO$_2$), and dissolved inorganic carbon (DIC), on the 19th June and 21st August, are shown in Fig. 3. In June, the seasonal stratification is well developed, with the surface layer warming and a surface-to-bottom temperature difference of 6 °C. A thermocline of approximate thickness 35 m separated the SML (depth 15 m) from the deep water (depth 50 m). The chlorophyll profile shows a pronounced maximum at around 35 m depth, signifying the SCM. Analysis of bottle samples reveal that within the SML there is negligible NOx whilst in the deep water the concentration is around 9 μ mol l$^{-1}$, implying a significant NOx gradient across the thermocline. The DIC concentration is also higher in the deep water with a DIC difference (△DIC) 70 μ molkg$^{-1}$ across the thermocline.

On the 19th June, there is a O$_2$ maximum within the thermocline of 273 m mol m$^{-3}$ (corresponding to oversaturation of > 105%). The O$_2$ concentration in the SML is 252 m mol m$^{-3}$, again representing an oversaturation (103%). As the SML is in direct contact with the atmosphere it is reasonable to assume that there is out-gasing of oxygen at this time. The O$_2$ concentration in the deep water is 262 m mol m$^{-3}$ showing an oxygen deficit (saturation 93%).

By the 21st August, the SML has warmed and deepened relative to June. The SCM is deeper and weaker than that observed in June whilst the NOx and DIC profiles are similar to those observed in June. Coincident with the deepening of the thermocline, the O$_2$ maximum

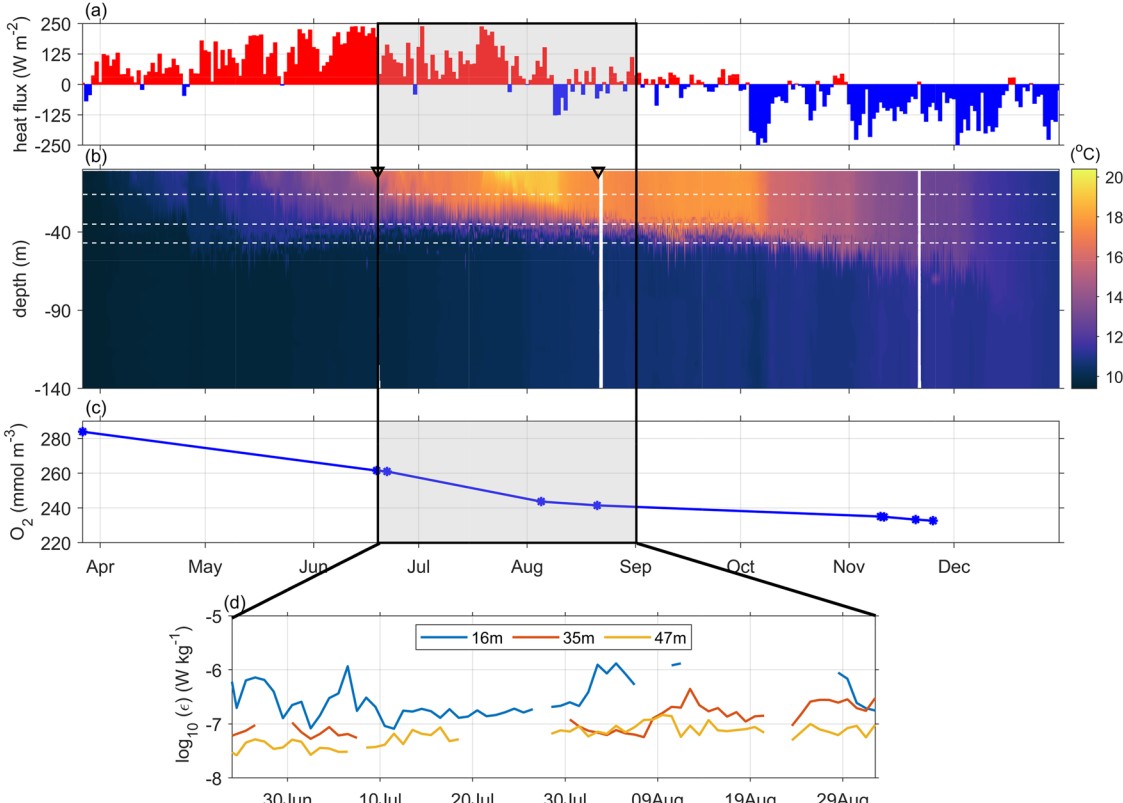

**Fig. 2 | Evolution of stratification, deep water oxygen concentration and turbulence at a seasonally stratified location in the central Celtic Sea (CTD on Fig. 1) over the summer of 2014. a** time series of net surface buoyancy forcing. **b** Evolution of water column temperature. **c** Deep water $O_2$ concentration. Each point is the average deep water dissolved oxygen concentration. **d** time series of the rate of dissipation of turbulent kinetic energy ($\epsilon$) at depths of 16 m (blue), 35 m (red), and 47 m (yellow). Source data are provided as a Source Data file.

is deeper relative to that observed in June. Again, there is an over-saturation of $O_2$ in the SML (101.5%) and within the subsurface oxygen maximum (109%). In the deep water, the oxygen concentration has declined by a further 20 m $mol$ m$^{-3}$, since the June profile, with a saturation at 87%. These changes imply a net rate of reduction in deep water oxygen concentration of 0.32 m $mol$ m$^{-3}$d$^{-1}$ between the 2 profiles.

For each profile, diapycnal fluxes for the biogeochemical parameters are estimated from the thermocline gradient and corresponding daily mean $\epsilon$ value using the dissipation method[28] (Table 1). The variability in the flux estimates is dominated by the variability in $\epsilon$ over the day for which the mean value is taken.

For June fluxes of DIC and NOx from the deep water into the thermocline are estimated to be $(20.8 \pm 5.4)$ m mol m$^{-2}$d$^{-1}$ and $(2.8 \pm 0.7)$ m mol m$^{-2}$d$^{-1}$ respectively. The respective gradients imply the flux has a C: N ratio of $7.3 \pm 1.5$, not significantly different from the C: N Redfield Ratio of 6.625[30]. Similarly, for August, the respective gradients imply a C: N ratio of $5.6 \pm 1.2$.

For both the June and August profiles, the thermocline $O_2$ maximum gives significant $O_2$ gradients between both the SML and the deep water. On 19th June, this results in $O_2$ fluxes to both the SML and deep water of 45.9 m mol m$^{-2}$d$^{-1}$ and $-4.0$ m mol m$^{-2}$d$^{-1}$, respectively. In August respective $O_2$ fluxes are 6.5 and $-20.2$ m mol m$^{-2}$d$^{-1}$.

Over the intervening period, the deep water $O_2$ concentration drops from 262 to 240 m mol m$^{-3}$. In flux terms this is equivalent to an $O_2$ removal rate of 36.0 m mol m$^{-2}$d$^{-1}$ (assuming a constant rate of decline and a deep water layer depth of 100 m). This estimate is between 2 and 10 times larger than the rate of supply by diapycnal mixing. All else being equal, the absence of diapycnal mixing of $O_2$ would imply a rate of decline of deep water $O_2$ of between

40 − 56 m mol m$^{-2}$d$^{-1}$. This is comparable to the estimated total $O_2$ flux from the mid-water $O_2$ maximum (50 and 27 m mol m$^{-2}$d$^{-1}$). Diapycnal mixing, therefore, significantly reduces the rate of development of the deep water $O_2$ deficit.

However, it is also estimated that 92% and 24%, respectively of the total $O_2$ fluxed out of the mid-water $O_2$ maximum goes into the SML. This upward $O_2$ flux will contribute to the observed 2−3% over-saturation along with any SML primary production.

### Diapcynal mixing mechanisms

In Fig. 4 we examine the potential contribution of the 2 main sources of mechanical mixing in seasonally stratified shelf seas: the tide[14,31,32] and the wind[17,18,33,34]. The variability in the two contributions is shown in Fig. 4a as a time series of the rate of dissipation of barotropic tidal energy and of wind energy.

The rate of barotropic tidal energy dissipation (Fig. 4a, blue) varies by an order of magnitude on semidiurnal timescales, and by a further order of magnitude on fortnightly spring-neap timescales. Periods of enhanced winds (Fig. 4a, red), particularly around the 10th of August and at the end of August, are intermittent lasting several days, and are associated with the passage of atmospheric low-pressure systems.

The mechanism for the generation of turbulence to support the mechanical mixing is shear instability. Within seasonally stratified shelf seas the thermocline tends to be marginally stable (eg. a gradient Richardson number, the ratio of stratification to the vertical shear in velocity, $(N^2/S^2) \approx 1$[16,24,33,35]). Accordingly, any significant shear enhancement will potentially reduce $N^2/S^2$ sufficiently to generate shear instability and mixing. The contributions to the vertical shear in the horizontal velocity thus provide an indicator for processes driving

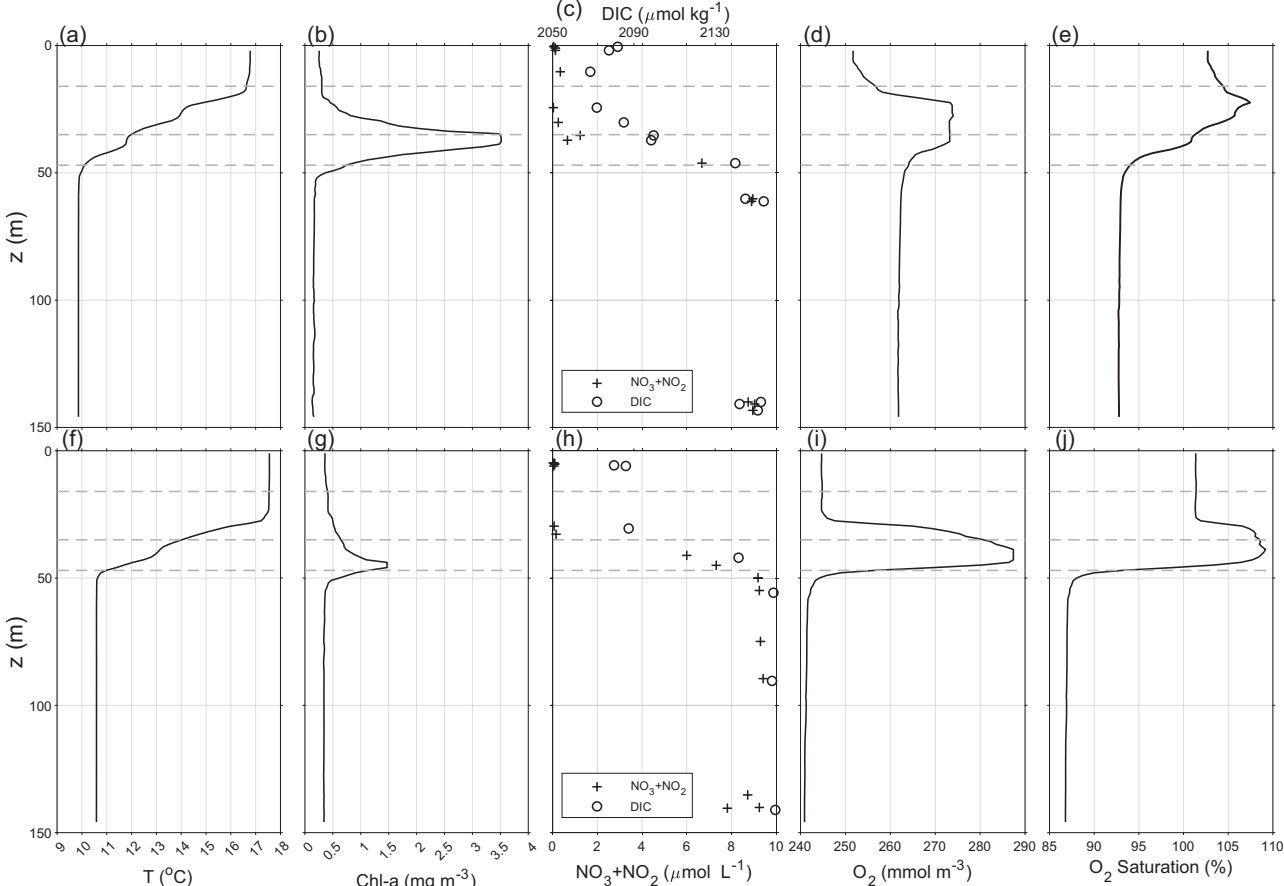

**Fig. 3 | Profiles of water column structure and distribution of dissolved matter.**
**a**, **f** temperature (°C), **b**, **g** chlorophyll ( mg m⁻³), **c**, **h** concentrations of dissolved
NOx (nitrate + nitrite) (black star) (μ moll⁻¹) and dissolved inorganic carbon (DIC)
(open circle) (μ molkg⁻¹), **d**, **i** concentration of dissolved oxygen ( m mol m⁻³) and

**e**, **j** oxygen saturation (%). Profiles **a**–**e** were collected on the 19th June and **f**–**j** on the
21st of August 2014. The dashed horizontal lines show the heights of the $\epsilon$ time
series. Source data are provided as a Source Data file.

mixing. Here the contributions of the two main sources of shear, the
(barotropic and internal) tide and wind (inertial), are separated spec-
trally for each of the three levels corresponding to the $\epsilon$ measurements
(Fig. 4b–d).

Overall, the magnitude of the shear decreases with depth, with the
weakest shear observed at 47 m, and shears approximately 10 times
and 50 times larger, at 35 m and 16 m respectively. At 16 m (Fig. 4b), the
shear is predominantly inertial (red line) and correlates with periods of
elevated wind forcing, while tidal contribution is weak, and only sig-
nificant during mid-July and in early August. At 35 m (Fig. 4c) both wind
and tide make a significant contribution to shear, with tidal shear

dominating during the 2 large springs tides (in mid-July and mid-
August). In contrast, the shear observed at 47 m (Fig. 4d) is dominated
by the tide and is strongly modulated by the 14-day spring-neap cycle
(blue line). There is no significant inertial shear evident at this depth
even during the periods of strong winds.

## Discussion

Coincident measurements of vertical profiles of biogeochemical para-
meters and time series of stratification and $\epsilon$ (a proxy for mixing) are
combined to estimate diapcynal fluxes, and to identify key processes
driving variability in deep water $O_2$, at a seasonally stratified shelf sea
location. The deep water is cut-off from direct contact with the atmo-
sphere by the seasonal thermocline, with the deep water $O_2$ decline a
consequence of usage (water column organic matter respiration and
sediment oxygen uptake) exceeding supply by diapcynal mixing[25,36,37].

Here we have shown that the upward $O_2$ flux from the mid-water
$O_2$ maximum into the SML, represents a significant proportion
(24–92%) of the total $O_2$ flux associated with the diapcynal mixing of
the mid-water $O_2$ maximum. As the SML $O_2$ concentration is over-
saturated it is reasonable to assume that the upward flux from the $O_2$
maximum is balanced by a sea surface flux to the atmosphere. Air-sea
$O_2$ fluxes are estimated using standard techniques[38] as 16 and
12 m $mol$ m⁻²d⁻¹ for June and August, respectively and are similar to
previous estimates for this region in summer[38]. They are also in rea-
sonable accord with the respective upward diapcynal flux estimates
reported here, which are based on water column measurements: 45.9
and 6.5 m mol m⁻²d⁻¹. These measurements therefore provide an

**Table 1 | Diapcynal flux estimates are based on the biogeo-
chemical profiles taken on the 19th of June and 21st of August
2014 combined with daily mean ε values at the appropriate
height (as indicated)**

| Date | Name (ε depth) | Flux |
|---|---|---|
| 19/06 | NO₂ + NO₃(36 m) | 2.8 ± 0.7 m mol m⁻²d⁻¹ |
| | DIC(36 m) | 20.8 ± 5.4 m mol m⁻²d⁻¹ |
| | O₂ up (16 m) | 45.9 ± 14.1 m mol m⁻²d⁻¹ |
| | O₂ down (47 m) | − 4.0 ± 1.2 m mol m⁻²d⁻¹ |
| 21/08 | NO₂ + NO₃(36 m) | 1.9 ± 0.5 m mol m⁻²d⁻¹ |
| | DIC(36 m) | 10.8 ± 2.8 m mol m⁻²d⁻¹ |
| | O₂ up (36 m) | 6.5 ± 1.7 m mol m⁻²d⁻¹ |
| | O₂ down (47 m) | − 20.2 ± 6.0 m mol m⁻²d⁻¹ |

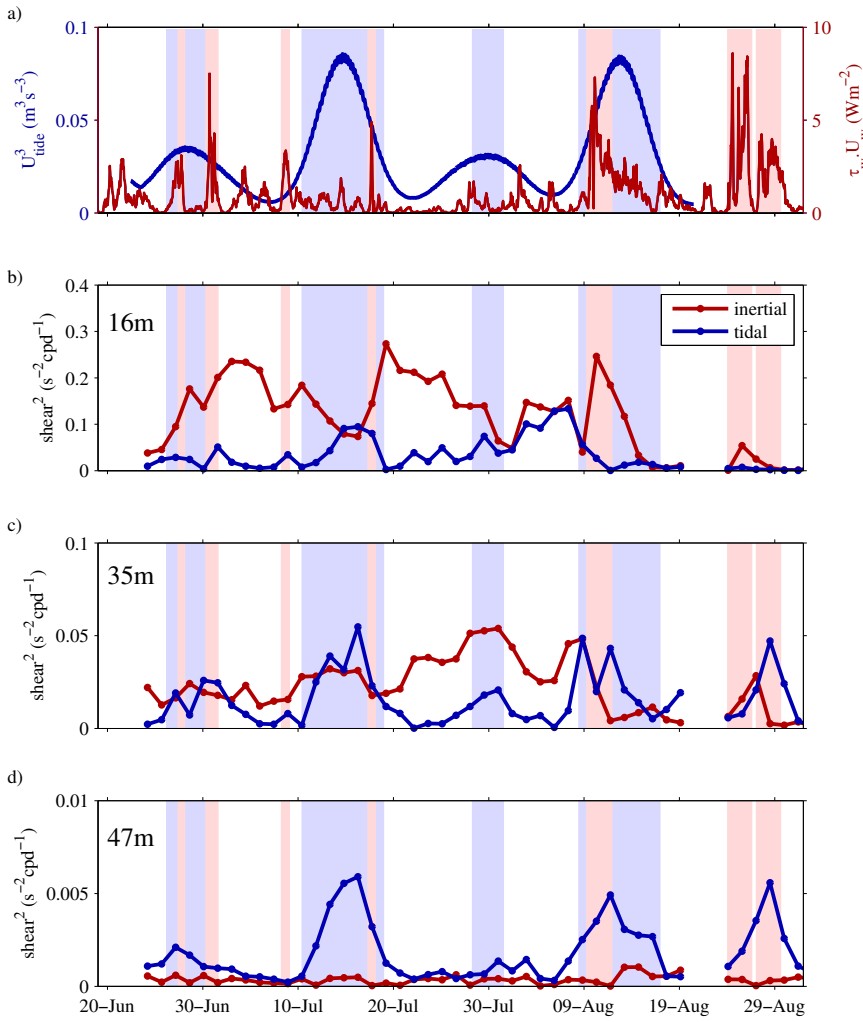

**Fig. 4 | Time series of wind and tidal energy input. a** The cube of the tidal current speed at 47m depth (blue) and the wind speed multiplied by wind stress (red). A 24 h running mean is applied to the tidal energy. The spectral amplitude of the vertical shear in the horizontal currents, squared, separated into tidal ($P_{m2S2} = 12.21$ h) and inertial ($P_I = 15.80$ h) frequencies, are plotted for 3.5-day windows at the depths of **b** 16 m **c** 35 m and **d** 47 m. Periods of; strong tides ($U^3 > 0.06$ m³s⁻³) are highlighted blue, and strong wind ($W_s\tau_s > 3$Wm⁻²) highlighted pink. Source data are provided as a Source Data file.

O₂ pathway from SCM primary production to sea surface outgasing. The processes involved are outlined in Fig. 5.

The observed average rate of decline in deep water O₂ is equivalent to a flux which is between 2 and 10 times larger than the downward diapcynal O₂ flux. Despite the uncertainty associated with both the assumption of a constant rate of deep water O₂ decline, and single profile estimates of the vertical fluxes, the result implies that the downward oxygen flux significantly compensates the O₂ loss due to respiration and remineralisation associated with sunken organic matter.

However, the apparent leakage of O₂ from the mid-water maximum into the SML implies that the SCM-deep water system is not closed. In consequence, the O₂ demand, generated by respiration and decay associated with the sunken and mixed down organic matter from the SCM, could exceed the O₂ supplied by diapycnal mixing, and so accelerate the net deep water O₂ decline.

The relative proportions of O₂ mixed upwards and downwards is dependent on the shape of the O₂ profile and the energy available for mixing. The profile shape is a legacy of buoyancy exchange across the sea surface, and both mid-water and boundary mixing. Spectral analysis of current shear across the thermocline suggests that whilst upward mixing of O₂ is largely associated with inertial shear, downward mixing is dominated by tidal processes. The relative magnitudes of the upward and downward O₂ fluxes is therefore determined by the

thermocline characteristics and the interplay between tidally induced shear and intermittent wind-driven inertial shear.

These results imply that the fate of the deep water O₂ in seasonally stratified seas, in a warming world, is linked to changing weather patterns impacting summer windiness, and in consequence, water column structure and diapcynal mixing. Climate change is also predicted to result in increasing seasonal stratification in these regimes[13] which could result in suppression of diapycnal mixing. However, the widespread observation of the marginally stable state of the seasonal thermocline[16,18,24,33,35] suggests that increased stratification may be offset by increased shear.

The planned major expansion of offshore wind capture into the seasonally stratified shelf seas will likely impact the development of the SML, SCM, and in consequence the seasonal deep water oxygen deficit. The move into deeper water requires a switch to floating turbine foundations. The tidal flow past these foundations will generate a turbulent wake which will provide an artificial source of diapycnal mixing in the upper part of the water column[8]. The impact of this additional mixing could be positive, for example, in shortening the period of stratification. However, these results highlight the need for the potential impacts of the modified diapycnal mixing to be considered in the design of turbine foundations and in the spatial planning of new wind farms.

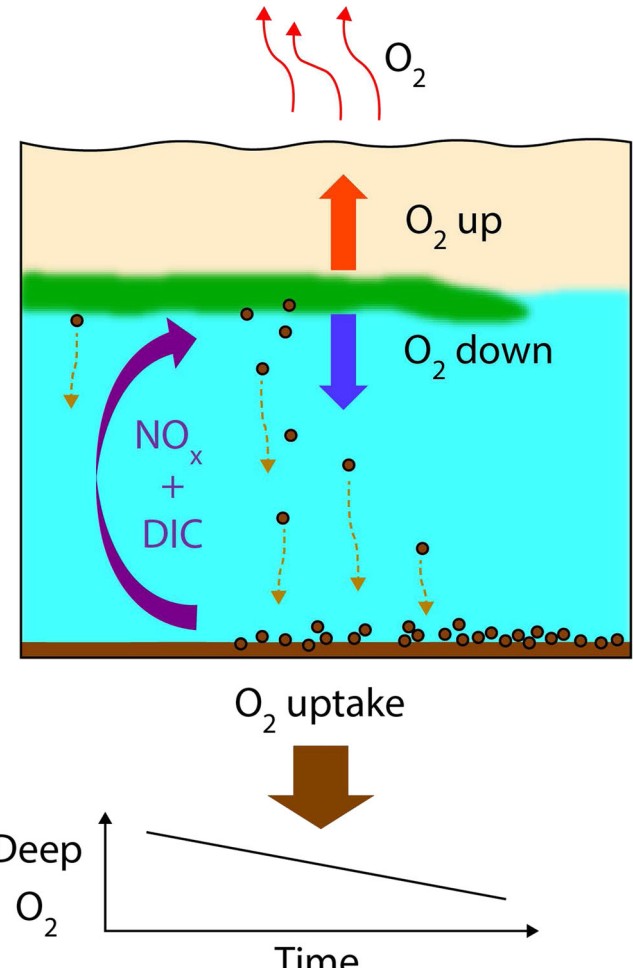

**Fig. 5 | Schematic illustrating the water column structure and diapycnal fluxes.**
A warm surface layer (pink) overlying cooler deep water (light blue) separated by a
thermocline. The mid-water green band indicates the position of the subsurface
chlorophyll maximum, and brown spheres represent sinking organic matter. The
fluxes due to diapycnal mixing are shown with dissolved inorganic carbon (DIC)
and limiting nutrients (NOx) as a purple arrow, downward $O_2$ flux as a blue arrow,
and upward $O_2$ flux as a red arrow. The oxygen usage associated with water column
respiration and sediment remineralisation of sunken organic matter is indicated by
the brown arrow at the seabed.

## Methods

### Water column structure timeseries and $\epsilon$ estimates

Estimates of turbulent kinetic energy dissipation rate ($\epsilon$) were derived
from velocity measurements made by three Teledyne RDI WorkHorse
Sentinel 600$kHz$ Acoustic Doppler Current Profiler (ADCP) instru-
ments mounted inline on a buoyancy tensioned mooring (location
shown in Fig. 1). The upper ADCP (S/N 7301) was installed upward-
looking in a syntactic buoy at a nominal depth of 23 m based on an
overall water depth of 145 m; the middle ADCP (S/N 3725) was
downward-looking in an open frame at a nominal depth of 36 m; and
the lower instrument (S/N 4015) was again upward-looking in a syn-
tactic buoy at a nominal 54 m[39]. Here we use data from two deploy-
ments spanning June 2014 to September 2014. The same instruments
were used for each deployment(The $\epsilon$ data are available from: https://
doi.org/10.17882/96076).

All three ADCPs had four-beam Janus style transducer heads, the
only difference between the instruments being that the upper and
middle instruments had a 20$^o$C beam angle (angle between beam and
along-instrument axis), whilst the lower ADCP had a 30$^o$ beam angle.
Each of the ADCP was configured in pulse-pulse coherent mode (RDI

mode 5), making single-ping ensemble (no averaging) observations of
along-beam velocity at 1Hz for 5 minutes, followed by 15 minutes sleep,
resulting in three bursts per hour, with each burst consisting of 300
profiles for each beam. The vertical resolution (bin size) for each ADCP
was 0.1 m, with the first bin centered at 0.97 m along the instrument
axis. The configuration gives an expected standard deviation for the
velocity estimates of 0.61$c$ ms$^{-1}$ with an anticipated profiling range of
3.5 m and a maximum relative water velocity of 1 ms$^{-1}$[40].

The Kolmogorov hypotheses[41] describe the second-order struc-
ture function, $D_{LL}(x,r) = <[\nu'(x+r) - \nu'(x)]^2>$, being the mean of the
square of the difference in turbulent velocity, $\nu'$, for separation dis-
tance $r$ relative to longitudinal position $x$ as being related to $\epsilon$ as:

$$D_{LL}(x,r) = C_2 \epsilon^{2/3} r^{2/3} \tag{1}$$

where $C_2$ is an empirical constant. A least-squares linear regression of
$D_{LL}$ over a range of separation distances using the model
$D_{LL}(x,r) = a_0 + a_1 r^{2/3}$ allows $\epsilon$ to be estimated as:

$$\epsilon = \left(\frac{a_1}{C_2}\right)^{3/2} \tag{2}$$

The turbulent velocity, $\nu'$, is typically isolated by adopting a Reynolds
decomposition of the observed velocity $v$ as $\nu = \bar{\nu} + \nu'$, where $\bar{\nu}$ is the
burst mean[42] or linear trend[43].

The presence of surface waves or the instrument heading oscil-
lating in a sheared flow results in periodic velocity gradients with
periods shorter than the burst duration. Consequently, non-turbulent
velocity differences are retained in $\nu'$ and will contribute to $D_{LL}(x,r)$,
resulting in a bias in the $\epsilon$ estimates. Scannell et al.[44,45] identify that such
periodic velocity gradients result in velocity differences that vary lin-
early with $r$ and hence their contribution to $D_{LL}(x,r)$ varies as $r^2$.
Adopting the alternative regression model:

$$D_{LL}(x,r) = a_0 + a_1 r^{2/3} + a_3 (r^{2/3})^3 \tag{3}$$

allows the non-turbulent contribution to be isolated from the
turbulent component which varies linearly with $r^{2/3}$, the modified
regression model coefficient $a_1$ again being used to calculate $\epsilon$.

Along-beam velocities were typically returned for bins 1 to 32 (1 to
29) for the 20$^o$(30$^o$)beam angle ADCP. Initial quality control rejected
values outside the range −1.1 ms$^{-1}$ to 1.2 ms$^{-1}$ considered to be affected
by phase-wrapping, as well as those with correlation values below 75
(scale 0 to 255)[46]. The echo intensity data was also used to exclude
velocities in accordance with the fish rejection algorithm, using the
default threshold[47]. Outlier values (exceeding three standard devia-
tions from the mean) over the burst and the beam profile were also
excluded. Since the modified regression model also isolates the $D_{LL}$
contribution due to linear shear, no detrending was applied to the
cleaned data.

The second-order structure function was calculated using a bin-
centred difference scheme, evaluated for separation distances of two
bins[45] and a least-squared regression using the modified model applied
to extract coefficients $a_0$, $a_1$ and $a_3$ for all instances where the
regression was possible. Instances where $a_3 < 0$; $a_0 < -1 \times 10^{-4}$ or
$a_0 > 3 \times 10^{-4}$; or the number of data points available for the regression
was less than eight were all excluded.

Daily mean $\epsilon$ estimates were calculated by taking the arithmetic
mean across the resolved bins in each beam, subject to a minimum of
six, and then across the resolved beams to give a burst mean. The daily
mean was the arithmetic mean of the resolved bursts, subject to a
requirement that at least 75% of the potential 72 bursts during the day
are resolved.

The time series of temperature and salinity were collected by 23
CTDs fixed to the same mooring deployment. Temperature and

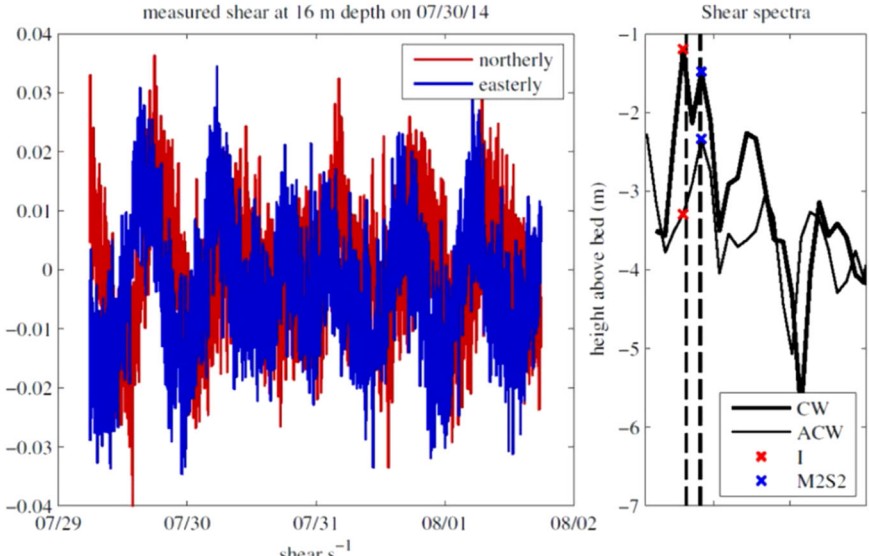

**Fig. 6 | An example of the shear and the spectral distribution of shear in the horizontal velocity.** The left-hand plot shows the measured shear at a depth of 16 m for the 30th July 2014. The right-hand plot shows the corresponding shear spectrum separated into the clockwise (thick line) and anticlockwise (thin line) components. The local inertial period (I) is indicated by the red crosses and the effective tidal period (M2S2) by blue crosses.

salinity data were collected with a temporal resolution of 5 minutes and a spatial resolution which varied with depth: 5 m (10 m − 35 m and 54 m − 79 m below the surface), 2 m (37 m − 49 m below the surface), and 10 m (89 m − 129 m below the surface)[48].

## Surface heat fluxes and wind

The time series of wind stress and direction at the research site were obtained from Met Office Ocean Data Acquisition Systems (ODAS) buoy data to. The wind was measured every 30 minutes and had been cross-checked the hourly ODAS buoy data with the 3-hourly data from European atmospheric reanalysis (ERA) satellite data. Surface stress and buoyancy flux were calculated using the TOGA COARE 3 bulk flux algorithm, taking account of the heights of the instruments on the ODAS buoy[49].

## Water Column Profiles

Profiles using a Seabird 991plus CTD on the 19th of June and 21st of August 2014 provided details of the water column structure at the site of interest. Output variables were extracted from the raw CTD data package by SBEDataProcessing software (Seasave Version 7.23.2).

CTD profiles were available from a series of research cruises during the UK Shelf Sea Biogeochemistry research programme (https://doi.org/10.1016/j.pocean.2019.102182). Vertical profiles were carried out with a Seabird 911Plus CTD, with data processed using Seasave version 7.23.2. Salinity was calibrated against IAPSO standard seawater to an uncertainty of ± 0.001($PSS$78). Chlorophyll concentrations (Chelsea Aquatracka MKIII) were calibrated against a laboratory standard to a typical uncertainty of ± 0.2 µgL$^{-1}$. Dissolved oxygen concentrations ($SBE$43) were calibrated via Winkler titration of triplicate water samples to an uncertainty of less than ± 0.2 µ molL$^{-1}$. In addition, water bottle samples were taken for dissolved inorganic carbon (at 12 levels) and nitrate+nitrite bottle (at 14 levels) between the sea surface and 150 m.

## Flux estimates

The vertical diffusivity rate ($K_z$) is derived from $N^2$ (estimated from the temperature and salinity timeseries data) and $\epsilon$ using the dissipation method[28]:

$$K_z = \Gamma\left(\frac{\epsilon}{N^2}\right)(m^2 s^{-1}) \tag{4}$$

where Γ is the dissipation flux coefficient:

$$\Gamma = \frac{R_f}{R_f + 1} \tag{5}$$

For the purposes of this paper we have assumed Γ = 0.2[50,51].

Heat and biogeochemical fluxes are then estimated by combining the $K_z$ estimate with an estimate of the appropriate diapycnal gradient across the point at which the $\epsilon$ measurement has been made. The temperature and $O_2$ gradients are estimated from the CTD profiles with the NOx (nitrate + nitrite) and dissolved inorganic carbon gradients (DIC) estimated from the discrete water bottle samples. For the latter two estimates of gradient are made, a direct fit across the pycnocline using the MATLAB 'gradient' function and a second based on the SML and BML averaged values and a thermocline thickness estimated based on the temperature profile[14,24]. In all cases, the two estimates were found to be consistent.

The fluxes were then calculated using:

$$J_{Flux} = - K_z \frac{\partial(NO_2 + NO_3)}{\partial z}(mmol\,m^{-2}s^{-1}) \tag{6}$$

$$J_{Flux} = - K_z \frac{\partial(O_2)}{\partial z}(mmol\,m^{-2}s^{-1}) \tag{7}$$

$$J_{Flux} = - K_z \frac{\partial(DIC)}{\partial z}(mmol\,m^{-2}s^{-1}) \tag{8}$$

The three main sources of uncertainty associated with this calculation are estimated and are due to variability in $\epsilon$ (up to 2 orders of magnitude over the averaging period of 1 day), variability in Γ (typically 20%) and uncertainty in the gradient estimate (typically ≈ 10%). The quoted values (Table 1) are dominated by sub-daily variability in $\epsilon$.

## Shear calculations

Vertical profiles of horizontal currents were collected at the mooring location over the period of interest using a bed-mounted Flowquest ADCP. Time series of vertical current shear (Fig. 4) were computed at the depth of each HF ADCP (16 m, 35 m, and 47 m) using bed-mounted Flow quest ADCP measured current velocities, averaged over the 10 m

above and below each instrument. Rotary power spectra were computed for 50% overlapping 3.5 – day sections of this data. This data window satisfies the Rayleigh criterion for the separation of frequencies, which are local inertial frequency ($I$ = 1.52 cpd) and effective tidal frequency ($m2S2average$ = 1.97cpd). The amplitude of the clockwise and anti-clockwise spectra, closest to these two frequencies was extracted from the spectra for each data window to give a time series of shear amplitude (CW + AC) at the inertial and tidal frequency for the duration of the deployment. Velocity data and example spectra are plotted in Fig. 6, showing the frequency of the inertial and semi-diurnal tide, and the closest spectral point to each in the clockwise and anticlockwise shear spectra.

### Air-sea flux calculations

The air-sea $O_2$ fluxes implied by the difference in $pO_2$ between the sea surface and the atmosphere are calculated following the methodology described by Brageron et al.[38] modified using Wanninkhof[52].

### Data availability

Source data are provided with this paper (https://doi.org/10.6084/m9.figshare.25459798.v1). The processed sea surface temperature data are available at https://data.neodaas.ac.uk/visualisation/. The processed epsilon data are available at https://doi.org/10.17882/96076. The processed CTD profile and bottle data in June are available at https://www.bodc.ac.uk/data/published_data_library/catalogue/10.5285/86532abd-d894-2c4c-e053-6c86abc0db01The processed CTD profile and bottle data in August are available at https://www.bodc.ac.uk/data/published_data_library/catalogue/10.5285/2eb8d803-8823-1e6f-e053-6c86abc052a6/ Source data are provided with this paper.

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

## Acknowledgements

The observations were collected as part of the United Kingdom (UK) Natural Environment Research Council (NERC) Carbon and Nutrient Dynamics and Fluxes over Shelf Systems (CaNDyFloSS) project, which forms part of the Shelf Sea Biogeochemistry research programme co-funded by the Department for Environment, Food and Rural Affairs (Defra) through UK Research and Innovation grant nos. NE/K001760/1 (TR), NE/K002007/1 (JS), NE/K001701/1 (JH), and NERC ENVISION DTP 1500369 (BS). We thank the officers and crews of the RRS James Cook and RRS Discovery, and the National Marine Facilities staff for their assistance in collecting the observations.

## Author contributions

T.P.R. planned and wrote the paper. S.S. carried out the flux calculations for her undergraduate research project. B.L. carried out the shear calculations. B.S. analysed the epsilon data. X.M. calculated the $O_2$ time series. J.H. led the deployment and recovery of the moored data used in the paper, and made a major contribution to the analysis of the data. J.S. led the project. S.S., B.L., B.S., X.M., J.S. and J.H. all made substantial contributions to the writing of this paper.

## Competing interests

The authors declare no competing interests.
