## [Peer Review File · Nature Communications]

The deep water oxygen deficit in stratified shallow seas is mediated by diapycnal mixingREVIEWER COMMENTS

Reviewer #1 (Remarks to the Author):

General comments:

In this manuscript, a mixing mechanism in the water column of seasonally stratified shelf seas is proposed that leads to additional reduction of oxygen saturation in the bottom waters during summer stratification. The peak of primary production in the thermocline (indicated by a strong sub-surface chlorophyll maximum) leads to upward fluxes of oxygen above the maximum, including outgassing to the atmosphere. This reduces the potential for downward oxygen fluxes towards the bottom and increases there the oxygen deficit due to mineralisation of organic matter.

This proposed mechanism is novel and relevant, and therefore the manuscript should be published in this journal. Some minor revisions would be necessary before.

As discussed below in more detail below, the argumentation could be formulated clearer, in wording as well as in improved tables and graphics. After reading the manuscript in depth, it is still not clear to me, if the authors simply claim that the bottom water oxygen saturation is further reduced simply due to outgassing of oxygen or if there are feedback mechanisms that further reduce the bottom oxygen concentration.

Detailed comments:

Abstract: It would be important to have clearer formulations of the impacts of the degassing of O₂ into the atmosphere, like in lines 74-75.

64-65: I do not fully understand this: „Accordingly, over the summer, diapycnal mixing promotes high carbon turnover in the deep water ...“ Would be nice to explain that by a few additional words. Does the carbon turnover profit from extra supply of oxygen? If so, this should be made clear.

74-75: Here the leakage as a cause for the O₂ deficit is clearly formulated. Would be important to also have such a clear formulation in the abstract.

Fig. 1: Would be good to mark the 17-deg C isotherm as a line.

95-112: I find this paragraph very hard to read, because it reports several results from dissipation observations at different time and different depths. This feels very descriptive. The authors should find a way to reformulate this paragraph, to give a clear message.

110-112: „... This data quality issues ...“, not clear what these „issues“ are, since observing dissipation rate inside the SML should not be an issue.

115-116: Maybe better „an average downward diapycnal heat flux of 40 W/m²“?

138-140: Wouldn't it be possible to estimate the oxygen flux to the atmosphere using bulk formulae as they are used in numerical models, depending on the wind speed and (maybe even the waves state)?

146: „there“ instead of „There“.

Figs. 3c&h: Here the symbols should get a legend within the panel, for better readability.

155-158: I guess these are the values for June, but you did not write it explicitly.

155-176: Although plausible, these numbers are not easy to digest. Would it be possible to additionally indicate these fluxes graphically, by some box plots with vertical arrows with size proportional to the fluxes? I know that space is limited in this journal, but such a graphics would help the reader tremendously. Maybe, table 1 (which could be much better organised) could be replaced by such a graphics.

190: Why do you define $N_2/S_2 = 1$ as the value of marginal stability? Theory says that it should be 0.25. When you decrease N_2/S_2 from 1.0 to 0.9, the water column should still be stably stratified. This seems to contradict what you write („any shear enhancement will potentially generate shear instability and mixing“).

Fig. 4a: It would be more consistent to plot that wind contribution as wind speed times wind stress. It would also be better to apply a 25-hour filter on the U3 values. Then, in both cases, the pink and blue areas would be more plausible.

226-230: Here the argumentation could be more clear. I understand that the decay of oxygen in the deep water due to usage for mineralisation is larger than the supply from diapycnal mixing. Therefore, the oxygen concentration in the deep water is going down. But in what sense does this accelerate the removal? Are you saying that we have a self-enhancing process here? How does that work?

Hans Burchard, 15.09.2023

Reviewer #2 (Remarks to the Author):

This is an excellent paper. It's very clear and straightforward despite the fact that it summarizes an extensive and diverse interdisciplinary data set. The main results are the clear demonstration that the mixing of DO downward from the pycnocline delays the decline in DO in the deepwater caused by respiration, and that the mixing flux is driven by interior shear driven by tides and augmented by near

inertial oscillations.

This paper should be read by all coastal marine scientists clarify the idea that low DO is due to reduced mixing because of summer increases in stratification.

The paper provides good evidence to support the claims and the methodology is sound.

I have only two suggestions: in the section where fluxes are described (Line 151), the text should refer to the methods for how the calculation was done: and since the rate of change of DO is based on the point samples data in figure 2c, the possibility that rates could be underestimated should be noted.

Nature Communications manuscript NCOMMS-23-40695-T: Response to reviewers.

We would like to thank the reviewers for their very helpful and positive comments. We hope we have been able to modify the manuscript to their satisfaction. Below we provide a line by line response to the reviewers comments together with details of the resulting modifications to the paper. Our response to the reviewers is highlighted for clarity.

Reviewer #1 (Remarks to the Author):

General comments:

In this manuscript, a mixing mechanism in the water column of seasonally stratified shelf seas is proposed that leads to additional reduction of oxygen saturation in the bottom waters during summer stratification. The peak of primary production in the thermocline (indicated by a strong sub-surface chlorophyll maximum) leads to upward fluxes of oxygen above the maximum, including outgassing to the atmosphere. This reduces the potential for downward oxygen fluxes towards the bottom and increases there the oxygen deficit due to mineralisation of organic matter.

This proposed mechanism is novel and relevant, and therefore the manuscript should be published in this journal. Some minor revisions would be necessary before.

As discussed below in more detail below, the argumentation could be formulated clearer, in wording as well as in improved tables and graphics. After reading the manuscript in depth, it is still not clear to me, if the authors simply claim that the bottom water oxygen saturation is further reduced simply due to outgassing of oxygen or if there are feedback mechanisms that further reduce the bottom oxygen concentration.

We are providing evidence to show that, due to the outgassing of some of the O₂ generated by the SCM primary productivity, the O₂ demands of the respiration and remineralisation associated with the sinking organic matter produced in the SCM will be greater than O₂ mixed down. Hence adding to the deep water O₂ decline. However, it's not a feedback mechanism. Hopefully the modification described below will clarify this further.

Detailed comments:

Abstract: It would be important to have clearer formulations of the impacts of the degassing of O₂ into the atmosphere, like in lines 74-75.

The middle sentences (lines 22-25) have been rewritten based on the reviewers feedback.

64-65: I do not fully understand this: „Accordingly, over the summer, diapycnal mixing promotes high carbon turnover in the deep water ...“ Would be nice to explain that by a few additional words. Does the carbon turnover profit from extra supply of oxygen? If so, this should be made clear.

These sentences have now been re-written (lines 64-67) for clarity.

74-75: Here the leakage as a cause for the O₂ deficit is clearly formulated. Would be important to also have such a clear formulation in the abstract.

See above

Fig. 1: Would be good to mark the 17-deg C isotherm as a line.

We have actually added the 16-deg C isotherm, as this provides a clear demarcation between seasonally stratified and mixed regimes in the Celtic Sea which is what we think the reviewer was looking for?

95-112: I find this paragraph very hard to read, because it reports several results from dissipation observations at different time and different depths. This feels very descriptive. The authors should find a way to reformulate this paragraph, to give a clear message.

We have rewritten this paragraph as requested (lines 95-110)

110-112: „... This data quality issues ...“, not clear what these „issues“ are, since observing dissipation rate inside the SML should not be an issue.

Yes – the reviewer is correct. This issue is not relevant to the narrative and so has been removed.

115-116: Maybe better „an average downward diapycnal heat flux of 40 W/m²“?

We agree – this sentence has been rewritten accordingly. (line 114)

138-140: Wouldn't it be possible to estimate the oxygen flux to the atmosphere using bulk formulae as they are used in numerical models, depending on the wind speed and (maybe even the waves state)?

Yes – we have now done this (lines:) with a description of the methodology added in the methodology section (lines:).

146: „there“ instead of „There“.

done

Figs. 3c&h: Here the symbols should get a legend within the panel, for better readability.

done

155-158: I guess these are the values for June, but you did not write it explicitly.

Yes – this is now stated. (line 253)

155-176: Although plausible, these numbers are not easy to digest. Would it be possible to additionally indicate these fluxes graphically, by some box plots with vertical arrows with size proportional to the fluxes? I know that space is limited in this journal, but such a graphics would help the reader tremendously. Maybe, table 1 (which could be much better organised) could be replaced by such a graphics.

We have now added a schematic (figure 5) which shows the fluxes. To help the reader further we have coloured the flux estimates in table 1 the same colour as the arrows in figure 5. As there are two separate flux estimates we felt it would be clearer to retain the table and colour code in this way rather than trying to include both sets of flux estimates on the schematic.

190: Why do you define $N^2/S^2 = 1$ as the value of marginal stability? Theory says that it should be 0.25. When you decrease N^2/S^2 from 1.0 to 0.9, the water column should still be stably stratified. This seems to contradict what you write („any shear enhancement will potentially generate shear instability and mixing“).

Okay – we have rewritten for clarity. See lines 188-194. The concept of marginal stability and generation of mixing by significant shear enhancement has previously been discussed in the references cited in this section.

Fig. 4a: It would be more consistent to plot that wind contribution as wind speed times wind stress. It would also be better to apply a 25-hour filter on the U3 values. Then, in both cases, the pink and blue areas would be more plausible.

Yes – plot 4(a) did actually have the wind contribution as wind speed times wind stress, however, the figure legend in the submitted draft was incorrect for which we apologise. As requested by the reviewer we have now plotted the tidal contribution having applied a 25-hour filter as requested by the reviewer.

226-230: Here the argumentation could be more clear. I understand that the decay of oxygen in the deep water due to usage for mineralisation is larger than the supply from diapycnal mixing. Therefore, the oxygen concentration in the deep water is going down. But in what sense does this accelerate the removal? Are you saying that we have a self-enhancing process here? How does that work?

We have rewritten the discussion to improve clarity. I think the key point in response to the reviewer, is that the mixing process is mixing less O₂ down from the SCM than is required to respire/remineralise the sinking organic matter associated with the primary production in the SCM (the magnitude of which is determined by the simultaneous mixing up of nutrients).

Hans Burchard, 15.09.2023

Reviewer #2 (Remarks to the Author):

This is an excellent paper. It's very clear and straightforward despite the fact that it summarizes an extensive and diverse interdisciplinary data set. The main results are the clear demonstration that the mixing of DO downward from the pycnocline delays the decline in DO in the deepwater caused by respiration, and that the mixing flux is driven by interior shear driven by tides and augmented by near inertial oscillations.

This paper should be read by all coastal marine scientists clarify the idea that low DO is due to reduced mixing because of summer increases in stratification.

The paper provides good evidence to support the claims and the methodology is sound.

I have only two suggestions: in the section where fluxes are described (Line 151), the text should refer to the methods for how the calculation was done:

We have now named the method used plus provided the relevant citation (full details of the flux calculations are provided in the methodology section 4.4). Lines 113-114 and 151-152.

since the rate of change of DO is based on the point samples data in figure 2c, the possibility that rates could be underestimated should be noted.

Good point, assumptions now stated in line 166 and then discussed in line 228-231.

We thank the reviewers and the editor for their time in considering this paper.

With Best Wishes,

Tom Rippeth on behalf of all of the authors.

REVIEWERS' COMMENTS

Reviewer #1 (Remarks to the Author):

The authors addressed all my concerns very well such that I do now agree with the other reviewer that this is a paper that should be read by all oceanographers who care about oxygen deficiency.

Hans Burchard (25.02.2024)